# Core Genome Multilocus Sequence Typing for Food Animal Source Attribution of Human *Campylobacter jejuni* Infections

**DOI:** 10.3390/pathogens9070532

**Published:** 2020-07-02

**Authors:** Chih-Hao Hsu, Lucas Harrison, Sampa Mukherjee, Errol Strain, Patrick McDermott, Qijing Zhang, Shaohua Zhao

**Affiliations:** 1Center for Veterinary Medicine, U.S. Food and Drug Administration, Laurel, MD 20708, USA; Chih-Hao.Hsu@fda.hhs.gov (C.-H.H.); Sampa.Mukherjee@fda.hhs.gov (S.M.); Errol.Strain@fda.hhs.gov (E.S.); Patrick.McDermott@fda.hhs.gov (P.M.); Shaohua.Zhao@fda.hhs.gov (S.Z.); 2College of Veterinary Medicine, Iowa State University, Ames, IA 50010, USA; zhang123@iastate.edu

**Keywords:** cgMLST, *Campylobacter jejuni*, source attribution, foodborne pathogens, PorA

## Abstract

*Campylobacter jejuni* is a major foodborne pathogen and common cause of bacterial enteritis worldwide. A total of 622 *C. jejuni* isolates recovered from food animals and retail meats in the United States through the National Antimicrobial Resistance Monitoring System between 2013 and 2017 were sequenced using an Illumina MiSeq. Sequences were combined with WGS data of 222 human isolates downloaded from NCBI and analyzed by core genome multilocus sequence typing (cgMLST) and traditional MLST. cgMLST allelic difference (AD) thresholds of 0, 5, 10, 25, 50, 100 and 200 identified 828, 734, 652, 543, 422, 298 and 197 cgMLST types among the 844 isolates, respectively, and traditional MLST identified 174 ST. The cgMLST scheme allowing an AD of 200 (cgMLST_200_) revealed strong correlation with MLST. cgMLST_200_ showed 40.5% retail chicken isolates, 56.5% swine, 77.4% dairy cattle and 78.9% beef cattle isolates shared cgMLST sequence type with human isolates. All ST-8 had the same cgMLST_200_ type (cgMLST_200_-12) and 74.3% of ST-8 and 75% cgMLST_200_-12 were confirmed as sheep abortion virulence clones by PorA analysis. Twenty-nine acquired resistance genes, including 21 alleles of *bla_OXA_*, *tetO*, *aph(3′)-IIIa*, *ant(6)-Ia*, *aadE*, *aad9*, *aph(2′)-Ig*, *aph(2′)-Ih*, *sat4* plus mutations in *gyrA*, 23SrRNA and L22 were identified. Resistance genotypes were strongly linked with cgMLST_200_ type for certain groups including 12/12 cgMLST_200_-510 with the A103V substitution in L22 and 10/11 cgMLST_200_-608 with the T86I GyrA substitution associated with macrolide and quinolone resistance, respectively. In summary, the cgMLST_200_ threshold scheme combined with resistance genotype information could provide an excellent subtyping scheme for source attribution of human *C. jejuni* infections.

## 1. Introduction

*Campylobacter* sp. are one of the leading causes of foodborne illness in the United States, causing an estimated 1.5 million infections each year [1,2]. Worldwide, campylobacteriosis is the most common cause of bacterial enteritis in humans with *C. jejuni* being the major contributor [3]. In otherwise healthy individuals, *C. jejuni* infections commonly result in a self-limiting diarrheal disease that is believed to commonly go unreported [4]. In immunocompromised patients, however, untreated *C. jejuni* infections can progress to sepsis and in rare cases meningitis [5,6]. Further, *C. jejuni* is associated with several sequelae including reactive arthritis, Guillain-Barre syndrome, Miller-Fisher syndrome and prolonged exposure in infants is associated with growth deficits [7,8,9]. Although considered a foodborne pathogen in humans, *C. jejuni* can be acquired from a wide range of sources including retail meats, raw milk, companion animals, fecal contamination, wild birds and other environmental reservoirs [10,11]. *C. jejuni* is also increasingly viewed as a reservoir for antibiotic resistance genes in both the environment and the food supply chain [2,12]. As a naturally competent organism, *C. jejuni* is capable of incorporating exogenous DNA during logarithmic growth [13]. This ability to readily acquire resistance genes from the environment enables *Campylobacter* strains to adapt to a range of antibiotic selective pressures found across multiple types of food animal facilities. *Campylobacter* species are also able to colonize food animals asymptomatically as a component of the gut microbiota and persist in the production environment with no apparent cost to the host [14,15].

Studies evaluating the prevalence of *Campylobacter* species in retail meats have shown higher rates in raw chicken (44%) compared to beef (0.5%) and pork (0.4%) sources [16]. While exposure to retail chicken meat is a major risk factor for campylobacteriosis, many isolates from chicken carcasses at food processing sites and retail chicken products do not show genetic relatedness to human pathogenic strains [17,18,19,20,21]. Strains recovered from beef cattle, dairy cattle, chickens, turkeys, swine, sheep and goats have all been linked to disease in humans [20,22]. This plurality of reservoirs demonstrates the need to evaluate the pathogenic contribution of each source to inform better practices and reduce the burden of campylobacteriosis on human health. The United States Department of Agriculture Food Safety and Inspection Service (USDA-FSIS) NARMS screening for foodborne pathogens in the ceca of several major food animals revealed prevalence of *Campylobacter* species in beef cattle (42%), dairy cattle (43%) and swine (32%) at much higher rates than found in retail meats, suggesting that infection from non-poultry animal sources may be underestimated [23]. The NARMS studies showed that high prevalence populations of *C. jejuni* were present in more food animal sources than previously observed. This broad range of food animal hosts combined with the potential for asymptomatic carriage and environmental contamination all confound efforts to identify the source of *Campylobacter* infections.

Many *C. jejuni* strains exhibit a narrow range of host specificity and this trait can be used to inform source attribution of strains that cause human infections [24,25]. However, strains with close genetic similarity can infect different hosts and established typing methods such as pulsed field gel electrophoresis (PFGE) or seven-gene multilocus sequence typing (MLST) lack the resolution to distinguish between them [10,18,26,27,28,29]. One proposed solution to this limitation is core genome fingerprinting, a method that expands on the seven loci of *C. jejuni* MLST to include 40 genomic targets conserved across *C. jejuni* genomes [29]. While this aids in strain differentiation, it also results in a labor-intensive set of five multiplexed PCR reactions. Analyses of the variable regions surrounding the *fla* gene or CRISPR loci have also been introduced, but these are often used in conjunction with MLST and incur a cost in labor [30].

In the last several years, whole genome sequencing (WGS) technology has been implemented in many public health surveillance programs as a fast and affordable alternative to PFGE and MLST [31,32]. Using WGS data for analysis can overcome the limitations of PFGE and MLST by evaluating the entire genome containing all housekeeping, virulence and resistance genes. Specifically, cgMLST typing schemes using a core genome of 1343 loci found in >95% of *C. jejuni* strains have demonstrated an increased discriminatory power over traditional MLST [33]. In a comparison of strains from multiple campylobacteriosis outbreaks, this cgMLST application was able to group strains from the same outbreak event into distinct clusters. We propose to explore the cgMLST approach further by determining whether the ability of cgMLST to establish genomic relatedness could be used to inform the source attribution of human pathogenic strains of *C. jejuni*. In the U.S., the contributions of PulseNet and the National Antimicrobial Resistance Monitoring System (NARMS) surveillance programs to the WGS repository maintained by NCBI provide an excellent source of foodborne pathogen sequence data against which to validate our cgMLST typing scheme. The objective of this study is to evaluate the utility of a cgMLST typing scheme for the source attribution of human pathogenic strains of *C. jejuni*.

## 2. Results

### 2.1. MLST and cgMLST Diversity Indices

The discriminatory power of cgMLST typing was determined by the number of AD between *C. jejuni* strains (Appendix A). We compared and evaluated traditional seven-gene MLST and cgMLST schema allowing for a range of 0 to 200 AD using Simpson’s index of diversity (Table 1). With regards to this dataset, a diversity index (D.I.) score of 1.000 indicated each sequence type contained no more than two strains while a D.I. lower than 1.000 represented a greater number of strains belonging to fewer conserved, groups. All cgMLST allelic difference thresholds evaluated resulted in a greater discriminatory power than traditional MLST. These results showed that the cgMLST scheme allowing an allelic difference of 200 (cgMLST_200_) had a discriminatory power most like traditional MLST with Simpson’s D.I. values of 0.964 vs. 0.963. The cgMLST_200_ schema classified the strains into 197 groups while traditional MLST generated 174 groups. Further, decreasing the allowable allelic differences generated a more stringent typing scheme resulting in an increasing number of highly related groups. It has been proposed that using AD less than 10 can be used for foodborne outbreak investigation [34].

### 2.2. Typing Schema and Source Attribution

To demonstrate the utility of the different typing methods for identifying the source of infection, typing groups from cgMLST and traditional MLST were evaluated by host origin. Specifically, cgMLST groups were classified by the property of containing strains isolated from humans. The ratio of isolates recovered from humans to isolates from each of the other sources was determined using cgMLST and traditional MLST (Figure 1). Traditional MLST and cgMLST_200_ typing methods showed similar representation of human pathogenic strains across all four food animal sources. The cgMLST_200_ scheme showed that 40.5% of isolates from retail chicken, 56.5% from swine, 77.4% from dairy cattle and 78.9% from beef cattle shared the same cgMLST type with isolates recovered from humans. In comparison, traditional MLST showed 53.6% of isolates from retail chicken, 56.5% from swine, 78.4% from dairy cattle and 81.7% from beef cattle were classified in human pathogenic typing groups. The prevalence of human pathogenic strains in these food animal sources is similar to previous analyses using a PFGE typing method [18]. Further, the similar contribution of human pathogenic strains between beef cattle and dairy cattle reflect a conserved composition of *C. jejuni* strains, as indicated by population analysis (Appendix A).

The similarities between traditional MLST and cgMLST were consistent with the diversity indices calculated earlier (Table 1). The stringency of grouping criteria (i.e., the threshold of differences between groups) reflected the number of isolates from non-human sources categorized into human pathogenic groups (Appendix A). Specific to groups containing isolates from humans, the cgMLST_200_ scheme identified 87 groups encompassing 404 strains, 222 of which were isolated from a human source (55%) while the more stringent cgMLST_5_ scheme identified 175 groups with 243 strains, 222 from a human source (91.4%). This trend of increased stringency favoring a more homologous source composition was conserved for all *C. jejuni* sources evaluated (Figure 2).

### 2.3. Validation of Typing Schema

One test of the utility of a typing scheme is its ability to categorize strains with phenotypically distinct traits into discrete groups. Traditional MLST has been shown to successfully categorize the sheep abortion (SA) strains of *C. jejuni* into ST-8 [35]. To validate the typing utility of the cgMLST scheme, we evaluated the distribution of strains with the SA-associated PorA sequence across both cgMLST_200_ and seven-gene MLST sequence types. All 74 strains identified as ST-8, as sequence type linked to the SA-associated PorA sequence, were classified into a single cgMLST_200_ group, cgMLST_200_-12. Within the entire population, 87/844 (10.3%) of the strains encoded for the SA-associated PorA sequence. The MLST-8 group contained 74 strains, 55 (74.3%) of which encoded the SA-associated PorA sequence (Table 2). The remaining 13 strains encoding for the SA-associated PorA sequence were found in ST 577, 61, 1244, 459 and 922 (Appendix A). The cgMLST_200_-12 group contained 76 strains, 57 of which (75%) had the SA-associated PorA, showing the cgMLST_200_-12 to be slightly better for classifying SA clones in this set of 844 *Campylobacter* genomes. Overlay of PorA annotation data onto a minimum spanning tree of cgMLST type revealed two clusters harboring the majority of strains encoding for the SA-associated PorA sequence: cgMLST_200_-12 and cgMLST_200_-50 (Appendix A). When the allelic difference threshold was decreased from 200 to 50, stronger associations between typing groups were observed. For example, 51/55 (92.7%) of cgMLST_50_-136, 9/10 (90%) of cgMLST_50_-15 and 6/7 (85.7%) cgMLST_50_-51 strains encoded for the SA-associated PorA sequence. The cgMLST_50_-136 group contained one of the isolates with the SA-associated PorA sequence not accounted for in the traditional ST-8 group. Together, these results indicate that cgMLST_200_-12 and ST-8 predominate but not exclusively harbor the SA-associated PorA variant. 

### 2.4. Antimicrobial Resistance Genotypes and cgMLST

A total of 29 unique antimicrobial resistance genes were identified in the 844 *Campylobacter* genomes, including 21 alleles of *bla_OXA_*, *tetO*, *aph(3′)-IIIa*, *aadE*, *aad9*, *aph(2′)-Ih*, *aph(2′)-h* and *sat4*. Antimicrobial resistance mutations and substitutions were observed among 271 strains, including GyrA at aa 68 (*n* = 168), 23SrRNA at bp 2074 (*n* = 3), 23SrRNA at bp 2075 (*n* = 8), 50SL22 at aa 103 (*n* = 112), RpsL at aa 43 (*n* = 2) and RpsL at aa 88 (*n* = 2). Certain cgMLST_200_ groups had conserved antimicrobial resistance genotypes. In total, 12/12 cgMLST_200_-4 and 6/6 cgMLST_200_-584 strains carried both *bla_OXA_*-193 and *tetO*, 12/12 cgMLST_200_-510 had the A103V substitution in L22, and 10/11 cgMLST_200_-608 the T86I substitution in GyrA. The *tetO* gene was also present in 74/76 cgMLST_200_-12 isolates. Within the population of 844 *Campylobacter* isolates, 636 encoded for a *bla_OXA_* gene. The most prevalent allele was *bla_OXA_*-193, present in 398 strains in all five host types evaluated. The remaining *bla_OXA_* alleles demonstrated specificity towards certain human sequence types. Of the *Campylobacter* strains isolated from humans, 10/10 cgMLST_200_-86 encoded *bla_OXA_*-61, 8/8 cgMLST_200_-510 encoded *bla_OXA_*-603, 5/5 cgMLST_200_-76 encoded *bla_OXA_*-449 and 4/4 cgMLST_200_-435 encoded *bla_OXA_*-447 (Appendix A).

The above antibiotic resistance patterns were then evaluated to determine if antibiotic resistance determinants could be used in conjunction with sequence type as an indicator of isolate source. The cgMLST_200_-4 group isolates were obtained from all host types evaluated, including beef, chicken, dairy, swine and human while the cgMLST_200_-584 group were obtained from either chicken or human. cgMLST_200_-510 isolates were obtained from chicken, swine and human sources and the cgMLST_200_-608 type was only found in chicken. Evaluating the *bla_OXA_* family of genes, only *bla_OXA_*-61 was able to help inform host source as no dairy cattle from that group encoded *bla_OXA_*-61. These data indicate that while antibiotic resistance determinants were restricted to sequence types, the distribution of alleles within a sequence type was not limited to a specific host.

## 3. Discussion

The cgMLST scheme described here provides a convenient strategy for strain typing by evaluating the 1343 loci present in the core genome of *C. jejuni*. This increased resolution over traditional seven-gene MLST allows cgMLST typing schemes to more accurately inform the source of human pathogenic *C. jejuni* strains. One factor that complicates the use of traditional MLST for source attribution is the ability of *C. jejuni* to infect multiple host types [36]. Using the cgMLST approach, we have demonstrated that allowing fewer allelic differences between isolates segregates strains from different sources into discrete groups (Figure 2). Further, source composition analysis of the typing groups reveals which sources harbor the greatest number of human pathogenic strains. In addition to reducing the number of potential sources, the allelic threshold informs the degree of genetic relatedness between human pathogenic strains and human non-pathogenic strains from known sources (Table 3).

Earlier approaches to increase the resolution of sequence typing methods have incorporated genomic determinants of antibiotic resistance [24,25]. In this study, 102 unique combinations of antibiotic resistance determinants were present in our population of *C. jejuni* isolates, with *tet(O)* and *bla_oxa_* alleles present in isolates obtained from each source evaluated (Appendix A). Our data revealed an interesting trend in antibiotic resistance between the food animal groups. Specifically, none of the determinants of macrolide resistance in the AMRFinder database were found in either beef or dairy cattle, food animal sources where macrolides are commonly administered by injection Although macrolides (e.g., tylosin) can apply selective pressure on *C. jejuni* during food animal production, the associated resistance determinants were absent in isolates recovered from the beef and dairy cattle populations in our study [37,38]. Interestingly, while two-thirds of the strains with macrolide resistance determinants were isolated from chicken sources, these same substitutions have been shown to incur a fitness cost and growth rate decrease in chickens in the absence of antibiotic selection pressure [39]. Strains lacking these substitutions have been shown to outcompete macrolide-resistant mutant strains [40,41]. In light of this, the absence of macrolide resistance determinants in isolates from our cattle populations compared to a 52.7% prevalence in our population of chicken isolates suggests that the distribution of macrolide resistance between these animal sources is not random.

One limitation of this study is that we do not have direct epidemiological evidence linking a pathogen from a specific source and sequence type to human disease. cgMLST shows strong genetic relatedness between human and non-human sourced isolates of *C. jejuni*; however, we currently lack the epidemiological metadata necessary to determine causality. We may conclude that there are strains of *C. jejuni* isolated from humans very closely related to strains isolated from food animal sources, differing only by the sequence of five alleles. A further limitation of the study is the underrepresentation of *C. jejuni* isolates from swine. While the sample size is too small to draw conclusions from our swine strains, we have demonstrated that a subset of isolates capable of infecting swine share a close genetic relatedness (i.e., AD < 5) with isolates that also infect humans, beef cattle, dairy cattle and chickens. The genomic similarity of these strains from different hosts suggests that these sequence types identify strains with a less restricted host range.

Source attribution of *C. jejuni* infections allows food industry professionals to tailor their practices to prevent the spread of foodborne pathogens. It also enables healthcare professionals to more efficiently use resources to contain the source of the disease. Similar to previous work, we used cgMLST to identify groups of genetically related strains of human pathogenic *C. jejuni* [33]. Expanding on this, we were also able to evaluate the cooccurrence of cgMLST groups, host source and antibiotic resistance determinants to identify which food sources harbor strains most closely related to pathogens that cause disease in humans. Our data demonstrate that evaluating Campylobacter strains across the 1343 loci of the cgMLST typing scheme provides a higher resolution method to determine genomic relatedness compared to traditional seven-gene MLST. Additionally, cgMLST typing has the potential to inform clinical decisions as the associations between cgMLST type and antibiotic resistance are established. Finally, as WGS data become more widely available, cgMLST typing serves as a cost-effective method for comparing multiple stains of *C. jejuni*. In conclusion, the cgMLST_200_ threshold combined with resistance genotype could provide an excellent subtyping scheme for source attribution of human *C. jejuni* infections.

## 4. Materials and Methods

### 4.1. Bacterial Strains and Sequencing

A total of 622 *C. jejuni* isolates recovered from food animals and retail meats were collected through NARMS starting in 2013 [42]. *C. jejuni* strains were obtained from the following sources: beef cattle cecal isolates (*n* = 213), dairy cattle cecal isolates (*n* = 181), chicken cecal isolates (*n* = 11), chicken retail meat isolates (*n* = 194) and swine cecal isolates (*n* = 23). Genomic DNA was extracted using the Qiagen DNeasy Blood and Tissue kit (Qiagen, Gaithersburg, MD) and genomes were sequenced on an Illumina MiSeq using v3 chemistry (Illumina, San Diego, CA). *C. jejuni* genomes were assembled using the de Brujin graph assembler from the CLC Genomics Workbench version 8.0 (CLC bio Aarhus, Denmark). These sequence data with a mean coverage of 51.37 were added to a collection of 222 sequences of human pathogenic *C. jejuni* that were submitted to the National Center for Biotechnology Information (NCBI) by the Centers for Disease Control and Prevention (CDC) to generate the pool of 844 genomes used in this project.

### 4.2. Traditional Multilocus Sequence Typing

The analysis of traditional seven-gene MLST was performed using whole-genome sequencing (WGS) data. *Campylobacter* multilocus sequence typing (MLST) allelic profiles and sequences were downloaded from the PubMLST database [43]. A total of 9098 profiles for seven different loci were used for the MLST analysis. The SRST2 pipeline was used to determine the MLST type for our *Campylobacter* isolates [44].

### 4.3. Core Genome MLST

The cgMLST scheme of 1343 loci, defined from 2742 *Campylobacter* isolates from Oxfordshire, UK, between 2011 and 2014, was downloaded from the PubMLST online database and queried against our collection of 844 *C. jejuni* using a BLAST search [33,43]. As the core genome represents genes present in >95% *C. jejuni* strains, up to 100 missing alleles were allowed for each genome [33]. The core-genome sequence type (cgST) for each isolate was determined based on the cgMLST scheme and a new cgST was assigned to a genome if the combination of alleles for the genome was not found in the cgMLST scheme. The cgST and allele composition for all genomes were assigned to the allelic profiles where missing alleles were represented with an “N”. Graphical representations of the population as minimum spanning trees from cgMLST data were generated using GrapeTree [45].

### 4.4. Hierarchical Clustering of cgMLST Data

Distance between genomes was calculated based on the pairwise distance between allelic profiles and was measured as allelic difference (AD). All genomes were clustered on seven different levels of AD: 0, 5, 10, 25, 50, 100 and 200 using the single-linage clustering method. With this method, a genome was linked to a group if the genome had less than or equal to the threshold number of AD with at least one other genome of that group. Genomes clustered together at each AD threshold were assigned the same group ID. 

### 4.5. Analysis of Population Composition

Underlying population composition was determined using STRUCTURE software v2.3.4 to analyze allele profiles across the 1343 loci of the *C. jejuni* core genome for the 844 strains present in this study. Software parameters were informed by prior source attribution studies and are described as follows [46]. The program was run using 50,000 burn-in iterations followed by 50,000 iterations applying a no-admixture model and assuming allele frequencies were independent. Source population was used to inform the analysis, however strains originating from humans were not used in model generation.

### 4.6. Identification of Antimicrobial Resistance Genotypes

Antimicrobial resistance genes were identified using the AMRFinder tool [47] that uses a curated AMR gene reference database of 4579 antimicrobial resistance proteins, more than 560 hidden Markov models (HMMs) and a curated protein family hierarchy. AMRFinder was also used to identify antibiotic resistance determinants in *Campylobacter*, including mutations and substitutions in the *gyrA* gene, 23S ribosomal RNA, 50S ribosomal protein L22 and 30S ribosomal protein S12 RpsL.

### 4.7. PorA Analysis

The sequence variation of the major outer membrane protein (PorA) is implicated in pathogenesis of *C. jejuni* [48]. The PorA sequence of the reference strain NCTC11168 was downloaded from NCBI GenBank as a wild-type comparator. This sequence was compared to our collection of 844 *C. jejuni* using an in-house perl script to determine the population-wide distribution of allelic variations of the loop 4 region in PorA, a region associated with a *C. jejuni* phenotype that can result in abortions in sheep. Isolates encoding a PorA sequence with less than 70% identity to the comparator strain NCTC11168 were excluded, leaving 739 *Campylobacter* isolates used in the PorA analysis. The loop 4 region of the sheep abortion (SA) associated PorA sequence was queried against the remaining *C. jejuni* sequences using blastx, and the results were aligned with ClustalW. The sequences were then visualized using MAFFT through EMBL to identify isolates with the SA-associated PorA sequence [49]. Sequence typing results from MLST and cgMLST were compared against the presence of the SA-associated PorA sequence to evaluate the ability of each typing scheme to identify SA strains of *C. jejuni*. 

## Figures and Tables

**Figure 1 pathogens-09-00532-f001:**
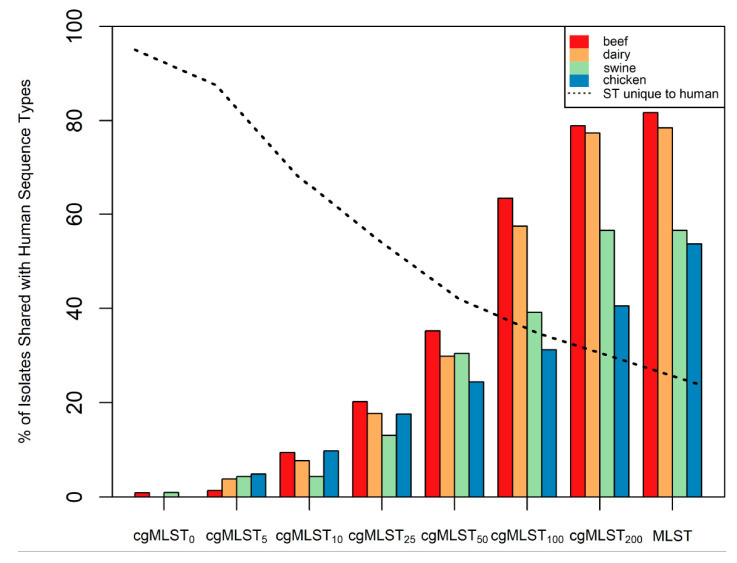
Proportion of *C. jejuni* from each source sharing a typing group with strains known to cause disease in humans. Column height indicates the percentage of strains from each source (beef, dairy, chicken or swine) that were classified in a group that contained human pathogenic strains of *C. jejuni*. The dashed line indicates the percentage of strains found in typing groups unique to humans.

**Figure 2 pathogens-09-00532-f002:**
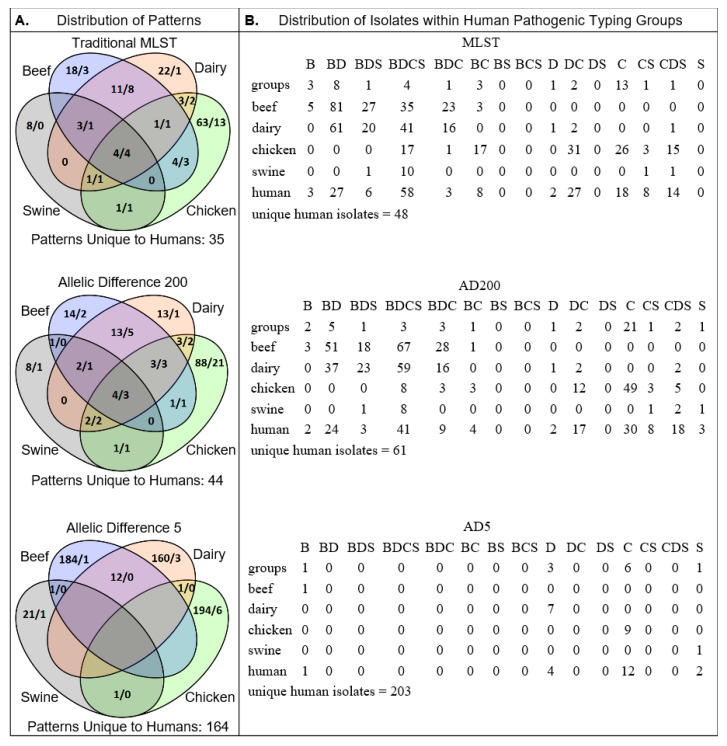
(**A**) Venn diagrams showing the number of cgMLST patterns associated with each source for traditional MLST (upper), cgMLST_200_ (middle) and cgMLST_5_ (lower). Patterns are numbered by total (left) followed by those shared with human pathogenic strains (right). (**B**) Distribution of individual strains belonging to the human pathogenic patterns in A. Isolates recovered from humans found in typing groups unique to humans are listed below for each typing scheme. Group totals correspond with A. and source identification is as follows: Beef (B), Dairy (D), Swine (S) and Chicken (C).

**Table 1 pathogens-09-00532-t001:** Comparison of MLST and cgMLST Typing Methods.

Typing Scheme	Schema Diversity		Number of Groups Generated				
	Number of Groups Identified *	Simpson’s D.I.	Beef	Chicken	Dairy	Human	Swine
MLST	174	0.963	33	54	34	64	16
cgMLST_200_	197	0.964	38	102	40	87	18
cgMLST_100_	298	0.983	73	131	68	103	22
cgMLST_50_	422	0.991	115	150	107	122	22
cgMLST_25_	543	0.997	149	168	141	145	23
cgMLST_10_	652	0.999	175	187	162	156	23
cgMLST_5_	734	0.999	196	196	173	175	23
cgMLST_0_	828	1	211	205	180	216	23

* Typing groups may be found in multiple hosts.

**Table 2 pathogens-09-00532-t002:** Alignment of PorA SA Determinant Region between Sequence Types.

	Sequence ^a,b,c^	cgMLST_200_-12 vs. MLST-8 ^d^	Description
170-	* * * * * * * * * * *	−209	Conserved residues
**NCTC11168**	FMAAEQGADLLEHSNISTTS NQAPFKVDSVGNLY		Reference sequence
**N48272F**	FMAEEQGADLLG**K**S**T** ISTT**QKA**APF**QA**DS**L**GNLY	55:53	SA clonal sequence
**N43804F**	FMA**E**EQGADLLG**K**S**T** ISIT **QKA**APF**QA**DSLGN**L**Y	1:1	Substitution not in SA loci
**N45191FR**	FMA**E**EQGT DLLG**K**S**T** ISTT**QKA**APF**QA**DS**L**GNLY	1:1	Substitution not in SA loci
**N46355F**	FMAEEQGADLLG**K**S**T**ISTT **QKA**APF**Q**TNS**L**GNLY	1:1	Substitution in SA loci
**N49694F**	FMEKEQ I S DLVG SNSSTFNVDSI GNLY	16:16	Deletion and substitutions in SA loci
**N44409F**	FMAAEQSS DLVG ANGSAFKVDSI ENLY	1:1	Deletion and substitutions in SA loci
**N45200F**	FMAKEQGSDLVG ANGSAFNVDSIGNLY	1:1	Deletion and substitutions in SA loci

^a^ * indicates conserved residue at sequence position, ^b^ Bold formatting indicates amino acid substitution associated with the SA phenotype, ^c^ Underline formatting indicates amino acid substitution not associated with the SA phenotype, ^d^ cgMLST200-12 (left) and MLST-8 (right) isolates identified with the corresponding PorA aa sequence.

**Table 3 pathogens-09-00532-t003:** Source Attribution of *C. jejuni* Isolates across Multiple Typing Schema.

Strain ID	MLST		cgMLST_200_		cgMLST_5_		cgMLST_0_	
	Group ID	Source *	Group ID	Source	Group ID	Source	Group ID	Source
**N46804F**	982	B,D,H	28	B,C,D,H,S	57	D,H	250	D
**SRR5217159**	982	B,D,H	28	B,C,D,H,S	57	D,H	754	H
**N45809F**	982	B,D,H	28	B,C,D,H,S	209	D,H	209	D
**SRR5878335**	982	B,D,H	28	B,C,D,H,S	209	D,H	826	H
**N48272F**	8	B,C,D,H,S	12	B,C,D,H,S	223	D,H	291	D
**SRR2970498**	8	B,C,D,H,S	12	B,C,D,H,S	223	D,H	528	H
**N50152F**	806	B,D,H,S	18	B,D,H,S	362	B,H	362	B
**SRR5604241**	806	B,D,H,S	18	B,D,H,S	362	B,H	806	H
**SRR5152202**	50	C,D,H	100	C,D,H	400	C,H	719	C
**SRR1794080**	50	C,D,H	100	C,D,H	400	C,H	400	H
**SRR2075414**	45	B,C,D,H,S	407	C,H	407	C,H	407	C
**SRR3029056**	45	B,C,D,H,S	407	C,H	407	C,H	407	H
**SRR2094236**	353	C,D,H	412	C,H	412	C,H	412	C
**SRR2182579**	353	C,D,H	412	C,H	412	C,H	444	H
**SRR3092138**	52	C,H,S	510	C,H,S	512	S,H	822	S
**SRR5754156**	52	C,H,S	510	C,H,S	512	S,H	551	H
**SRR5152177**	2083	C,H	584	C,H	593	C,H	714	C
**SRR5932391**	2083	C,H	584	C,H	593	C,H	833	H

* Beef (B), Dairy (D), Human (H), Chicken (C) and Swine (S).

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
