# Peer review of "Core Genome Multilocus Sequence Typing for Food Animal Source Attribution of Human *Campylobacter jejuni* Infections"

_pathogens, 2020, doi:10.3390/pathogens9070532_

Round 1

Reviewer 1 Report

Manuscript Number: 824271

Manuscript Title: Core Genome Multilocus Sequence Typing for Food Animal Source Attribution of Human Campylobacter jejuni Infections

Comments:

The cgMLST200 threshold scheme described in this interesting work, combined with resistance genotype information can contribute to provide a useful subtyping scheme for source attribution of human C. jejuni infections, even if epidemiological investigations are fundamental in this type of analysis, especially in cases of tracing outbreaks. The authors have shown very well how the allelic threshold informs the degree of genetic relatedness between human pathogenic strains and human non-pathogenic strains from known sources. The topic is of great interest to the public health community and worth attention. The data of the manuscript is solid.

I have enjoyed your research paper that will provide valuable information on the molecular determinants combined to cgMLST200.

Please consider my suggestions below.

Line 43-45: please lists the others sources of human campylobacteriosis. Those reported in the text are too few.

Line 46: not only “within the food supply chain”, but also in the environment. Pleas add.

Line 50: Campylobacter are part of the microbiota gut of many food animals. Please add this consideration.

Line 57: Please add this reference “Di Giannatale E, Garofolo G, Alessiani A, et al. Tracing Back Clinical Campylobacter jejuni in the Northwest of Italy and Assessing Their Potential Source. Front Microbiol. 2016;7:887. Published 2016 Jun 13. doi:10.3389/fmicb.2016.00887”

Lines 115-118: The indicated percentages, where are shown? I do not see these % in the figure 1. Please specify better in the text describing the figure 1 with more details.

In the table 1, what does the symbol #  refer to?

Line 135: what does the asterisk refer to?

Figure 2: what does the symbol #  refer to?

The three figures in the panel (A) are too close. Please distribute them better to make them clearer

Line 147: indicates the extended name followed by the abbreviation

Line 203: please delete the hypen before Beef

Line 217: please add comments on the other antimicrobials (tetracycline, aminoglycosides, beta-lactams…) whose resistance genes were found in the study.

Line 237-238: Please, explain better this sentence.

Line 252: please indicate the year for human isolates downloaded. In addition, if possible, it would also be interesting to indicate the sampling sites and discuss the importance of the epidemiological investigations combined with WGS analysis.

Thank you for considering my suggestions!

Author Response

Reviewer #1 Comments

Line 43-45: please lists the others sources of human campylobacteriosis. Those reported in the text are too few.

We agree with the reviewer that the range of sources of human campylobacteriosis presented was too limited.  We have broadened the scope to include companion animals and fecal contamination as additional sources in line 44.

Line 46: not only “within the food supply chain”, but also in the environment. Pleas add.

We agree that environmental sources certainly act as reservoirs for antibiotic resistance genes and we have added this consideration to the text in line 46.

Line 50: Campylobacter are part of the microbiota gut of many food animals. Please add this consideration.

We agree that Campylobacter should be recognized as a microbiotic component of many food animals and have included this along with an appropriate citation in line 51.

Line 57: Please add this reference “Di Giannatale E, Garofolo G, Alessiani A, et al. Tracing Back Clinical Campylobacter jejuni in the Northwest of Italy and Assessing Their Potential Source. Front Microbiol. 2016;7:887. Published 2016 Jun 13. doi:10.3389/fmicb.2016.00887”

We recognize the contribution of this reference and have included it in the text in line 57.

Lines 115-118: The indicated percentages, where are shown? I do not see these % in the figure 1. Please specify better in the text describing the figure 1 with more details.

We recognize that a mis-labeling error in Figure 1 where swine was labeled as chicken would have made it difficult to identify the correct percentages from the bar graph.  The figure has been updated and the column height corresponds to the correct percentage for each food animal group.  Figure 1.

In the table 1, what does the symbol #  refer to?

We recognize that the # symbol is not a universal convention for number and have altered the figure text for clarity.  Table 1.

Line 135: what does the asterisk refer to?

The asterisk is referred to in the table footnote as an indication that typing groups may be found in multiple hosts.  This is to explain why the sum of typing groups across all hosts is greater than the total number of typing groups that were identified.  We have adjusted the position of this footnote to make its association with the table more easily recognizable.  Line 136.

Figure 2: what does the symbol #  refer to?

We recognize that the # symbol is not a universal convention for number and have altered the figure text for clarity.  Figure 2.

The three figures in the panel (A) are too close. Please distribute them better to make them clearer

We agree that there was insufficient spacing between the figures and have adjusted the figure accordingly.  Figure 2.

Line 147: indicates the extended name followed by the abbreviation

We agree that SA needs to be defined and have extended it to read “sheep abortion”, followed by the abbreviated (SA) in line 148.

Line 203: please delete the hypen before Beef

We agree that the hyphen separating Beef from the asterisk is unnecessary and have removed it from the manuscript. Line 205.

Line 217: please add comments on the other antimicrobials (tetracycline, aminoglycosides, beta-lactams…) whose resistance genes were found in the study.

We agree with the reviewer’s comments that additional information on antibiotic resistance trends among the C. jejuni isolates would add value to the manuscript.  While the distribution of macrolide resistance determinants was the most prominent trend, we also recognize that the breadth of antibiotic resistance determinants present in the sample population is useful information and have included a summary table of antibiotic resistance determinants by source as well as the following sentence in lines 207 – 210:

“In this study, 102 unique combinations of antibiotic resistance determinants were present in our population of C. jejuni isolates, with tet(O) and blaOXA alleles present in isolates obtained from each source evaluated (Table S3).”

Line 237-238: Please, explain better this sentence.

To add clarity to this sentence, we have changed it to: "Our data demonstrate that evaluating Campylobacter strains across the 1,343 loci of the cgMLST typing scheme provides a higher resolution method to determine genetic relatedness when compared to traditional 7-gene MLST." Lines 240-243

Line 252: please indicate the year for human isolates downloaded. In addition, if possible, it would also be interesting to indicate the sampling sites and discuss the importance of the epidemiological investigations combined with WGS analysis.

We agree with the reviewer that this is important information; all sequences for C. jejuni isolated from humans were downloaded in 2016, and we have added this information to the text.

Unfortunately, we do not have sufficient data to run a comparative analysis of sample site, human disease and sequence type.  For example, while we are able to pair our food animal data to a specific state, all human isolates are simply designated as 'USA', preventing a geographical comparison between the groups.  Additionally, the 622 non-human source isolates are represented by 4 food animal sources across 45 locations.  This provided too few strains per food animal per location to draw any meaningful conclusions.  With a greater sample size and overlay of meat distribution patterns, however, we agree that this would be a very interesting analysis to perform.

Reviewer 2 Report

This articles presents an interesting core genome analysis of C. jejuni isolates from NARMS.

Major comments:

The authors should not use the term source attribution unless they conduct a formal statistical modelling exercise as detailed in https://www.ncbi.nlm.nih.gov/pmc/articles/PMC6820127/. 

While I agree that their cgMLST200 scheme is equivalent to 7-gene MLST, the 7-gene MLST scheme still remains the most used typing system and nomenclature for groups which don't have access to the cgMLST200 scheme. I would therefore recommend to replace the cgMLST200 groups (e.g. in abstract, lines 171-186 with the 7-gene MLST type, to make the results more informative to other readers)

In the supplementary table S1, it is not clear what ST types with a dash mean, e.g. what is the difference between ST21 & ST21-1.

The authors should submit any novel MLST types to Pubmlst for MLST type assignation.

It would be informative to produce minimum spanning trees of: 

1) most frequently occurring MLST types to check if isolates cluster by source

2)  For ST-8 to check whether SA variants cluster by source. what was the ST of the 13 SA isolates which were not ST-8, a single locus variant of ST-8?

3) A table is missing showing antibiotic resistance mechanisms by source. This is mentioned in discussion (e.g. line 211) without full data being provided.

4) Table 3 should be part of results somehow, not the discussion.

5) There is a curated nomenclature available for porA (https://pubmlst.org/bigsdb?db=pubmlst_campylobacter_seqdef&page=downloadAlleles&tree=1)

Please add the equivalent PorA alleles from these isolates to Table 2.

Minor points:

1) define what SA means

2)  Add how  raw data  be obtained to methods section.

3) Add what average coverage was in Materials and methods.

4) What Assembler does CLC Genomics use?

5) line 304: title is missing for Table S1

6) Provide time and location of strain collection in abstract.

7) line 79 typo MSLT, 

8) line 85 add able "to" group

9 line 91 add evaluate "the" utility

Author Response

Reviewer #2 Comments

Major comments:

The authors should not use the term source attribution unless they conduct a formal statistical modelling exercise as detailed in https://www.ncbi.nlm.nih.gov/pmc/articles/PMC6820127/.

We agree with the reviewer that it is important to incorporate one of the source attribution statistical modeling exercises from the article listed.  To address this, we have performed an analysis of population cluster composition of the food animal sources using the STRUCTURE software v2.3.4.

While I agree that their cgMLST200 scheme is equivalent to 7-gene MLST, the 7-gene MLST scheme still remains the most used typing system and nomenclature for groups which don't have access to the cgMLST200 scheme. I would therefore recommend to replace the cgMLST200 groups (e.g. in abstract, lines 171-186 with the 7-gene MLST type, to make the results more informative to other readers)

We agree that the 7-gene MLST naming scheme is more readily recognizable than the cgMLST200 scheme presented here.  Unfortunately, while the 7-gene MLST scheme and cgMLST categorize strains into similar groups, these groups are not identical.  Because of this we cannot substitute the most similar MLST group name in the analysis of the cgMLST200 groups as they can contain strains with different AMR profiles.

In the supplementary table S1, it is not clear what ST types with a dash mean, e.g. what is the difference between ST21 & ST21-1.

We thank the reviewer for recognizing this as these dashes are an artifact of in-house analysis indicating the initial sequencing runs resulted in low-quality data at the cgMLST loci and that these strains required re-sequencing prior to analysis.  Extraneous dashes and numeric designations have been removed.

The authors should submit any novel MLST types to Pubmlst for MLST type assignation.

It would be informative to produce minimum spanning trees of:

1) most frequently occurring MLST types to check if isolates cluster by source

We appreciate this comment by the reviewer and have generated MSTs for the population based on cgMLST profiles and annotated by source, typing group and presence of the SA-associated PorA sequence.

2)  For ST-8 to check whether SA variants cluster by source. what was the ST of the 13 SA isolates which were not ST-8, a single locus variant of ST-8?

We appreciate and agree with the reviewer's concern that the SA variants may be closely-related neighbors of the traditional ST-8.  To address this, we have included a table of the MLST profiles from strains containing the SA associated PorA sequence that demonstrates all associated ST's differ from ST-8 at 2 or more of the 7 MLST loci (table S2).  Additionally, we have included an overlay of PorA annotation to the MST referred to above.

3) A table is missing showing antibiotic resistance mechanisms by source. This is mentioned in discussion (e.g. line 211) without full data being provided.

We agree that the data provided in table S1 was not optimized to communicate the prevalence of antibiotic resistance determinant by source.  To facilitate this, we have generated an additional summary of results as Table S3 that details how the antibiotic resistance determinants are distributed across isolates recovered from humans and food animals.

4) Table 3 should be part of results somehow, not the discussion.

We agree with the reviewer’s comments that tabular reporting of data is most often presented in the Results section, and that if the purpose of the paper was to identify these 9 C. jejuni strains that were closely related to human-pathogenic strains then this table would certainly belong in the Results. However, this table is presented as a template that can be applied to other datasets to compare genomic relatedness between C. jejuni strains.  As this demonstrates the utility of the cgMLST typing scheme for use in the broader scientific community instead of a directed analysis of the data we obtained, we believe this table is more appropriately an element of the Discussion.

5) There is a curated nomenclature available for porA (https://pubmlst.org/bigsdb?db=pubmlst_campylobacter_seqdef&page=downloadAlleles&tree=1)

Please add the equivalent PorA alleles from these isolates to Table 2.

We agree with the reviewer that a uniform naming scheme is essential for clear scientific communication.  However, our PorA analysis was dependent upon the protein sequence detailed by Zhang et al. whereas the naming scheme of porA in the cgMLST at loci 1178 is a nucleotide sequence.  While in most cases we can match the PorA protein sequence in our dataset to a cognate porA nucleotide sequence from PubMLST using a tBLASTn query, the PorA sequences found in strains N45191FR and N46355F are not encoded for by any of the porA alleles in the PubMLST database.  For this reason, we would like to maintain a consistent naming scheme within the manuscript and retain the original labeling of the table.

Minor points:

1) define what SA means

We agree with the reviewer that SA needs to be defined in-text.  The SA designation refers to a virulent clonal type known to cause abortion in sheep.  The abbreviation is now defined in line 147.

2) Add how  raw data  be obtained to methods section.

We agree with the reviewer's comment that transparency of methods is essential.  However, we are unsure as to which step in the raw data acquisition process was not sufficiently described.  In the interest of clarity, we have included a reference to the USDA-FSIS document detailing sample acquisition from the food animal sources.

3) Add what average coverage was in Materials and methods.

We agree with the reviewer that sufficient read coverage must be achieved to ensure the validity of sequence data.  To address this, we have included in the text that the mean coverage was 51.37 reads per base-pair in line 257.

4) What Assembler does CLC Genomics use?

We recognize the reviewer's concern that different assemblers are better suited to certain assembly tasks than others.  CLC Genomics does not cite the use of any of the common de novo assemblers (e.g. SPAdes, SOAPdenovo, SKESA, SGA, Canu or even Velvet). Their whitepages cite use of a de Bruijn graph assembly algorithm (like SPAdes, etc.), which is an approach many assemblers employ when processing short read data.

5) line 304: title is missing for Table S1

We appreciate the reviewer spotting this and we have added a title to this and the subsequent supplemental materials.

6) Provide time and location of strain collection in abstract.

We agree with the reviewer’s concern that time and location data are important when making comparisons between strains and have included this information in the abstract.

7) line 79 typo MSLT,

8) line 85 add able "to" group

9 line 91 add evaluate "the" utility

We most certainly appreciate the reviewer’s attention to detail and have made the necessary corrections to points 7, 8 and 9.